# Papillomavirus DNA is not Amplifiable from Bladder, Lung, or Mammary Gland Cancers in Dogs or Cats

**DOI:** 10.3390/ani9090668

**Published:** 2019-09-08

**Authors:** John S. Munday, Chloe B. MacLachlan, Matthew R. Perrott, Danielle Aberdein

**Affiliations:** 1School of Veterinary Science, Massey University, Palmerston North 4410, New Zealand (C.B.M.), (M.R.P.), (D.A.); 2Carlyle Vet Clinic, Napier 4104, New Zealand

**Keywords:** papillomavirus, cancer, lung, mammary gland, bladder, dog, cat, viral carcinogenesis, feline papillomavirus, canine papillomavirus

## Abstract

**Simple Summary:**

Papillomaviruses (PVs) are well established to cause genital and oral cancer in humans. Additionally, some evidence suggests that these viruses may also have a role in the development of human lung, breast, and bladder cancer. Recent studies have revealed that PVs can influence the development of some cancers in cats and, less frequently, in dogs. In the present study, 47 lung, mammary gland, and bladder cancers in dogs and 25 cancers in cats were evaluated for evidence of any role PVs may have in cancer development. Histology did not reveal any lesions suggestive of PV infection, and no PV DNA was amplified from any cancer. Therefore, these findings suggest that PVs do not frequently infect these tissues and are therefore unlikely to be significant factors in the development of lung, mammary gland, or bladder cancer in either dogs or cats.

**Abstract:**

Papillomaviruses (PVs) cause around 5% of all human cancers, including most cervical cancers and around a quarter of all oral cancers. Additionally, some studies have suggested that PVs could cause a proportion of human lung, breast, and bladder cancers. As PVs have been associated with skin cancer in cats and, more rarely, dogs, it was hypothesized that these viruses could also contribute to epithelial cancers of the lung, mammary gland, and bladder of dogs and cats. Formalin-fixed paraffin-embedded samples of 47 canine and 25 feline cancers were examined histologically for evidence of PV infection. Additionally, three sets of consensus PCR primers were used to amplify PV DNA from the samples. No histological evidence of PV infection was visible in any of the cancers. DNA from a bovine PV type was amplified from one sample, while two different samples were found to contain human PV DNA. However, these were considered to be contaminants, and no canine or feline PV types were amplified from any of the cancers. These results suggest that PVs do not frequently infect the lung, mammary gland, or bladder of dogs and cats and therefore are unlikely to be significant factors in the development of cancers in these tissues.

## 1. Introduction

Papillomaviruses (PVs) infect the stratified epithelium of all species that have been intensively studied [1]. While the overwhelming majority of PV infections are asymptomatic [2], PVs can influence cell growth and division, and some PV infections can cause self-resolving hyperplastic papillomas—warts [3]. Additionally, PVs can cause neoplasia with around 5% of all human cancers thought to be caused by PV infection [4]. In dogs, 19 different PV types are recognized [5]. However, PVs are currently not thought to be common causes of cancer in this species with only sporadic reports of PVs causing oral or cutaneous neoplasia [3,6]. Five PV types have been identified in cats, with FcaPV-2 thought to be the predominant cause of pre-neoplastic Bowenoid in situ carcinomas (BISCs) and up to 20% of cutaneous squamous cell carcinomas (SCCs) in this species [7,8,9]. There is no evidence that PVs are a significant cause of oral cancer in cats [10,11].

While the role of PVs in genital, anal, and oral cancers in humans has been extensively studied and is well established [12], whether or not these viruses influence the development of cancers elsewhere in the body is less certain. A potential role of PVs in the development of human epithelial lung cancer was first suggested in 1979 [13]. Since then, studies have reported detection rates of PV DNA in lung cancers that have varied from 0% to 80% [14]. The majority of studies report the detection of ‘high-risk’ human papillomavirus (HPV) DNA in around a third of samples, although HPV DNA was also detected in non-neoplastic tissue in many of these studies [14]. Therefore, while there is evidence to suggest a potential role of PVs in human lung cancer, the role of these viruses is unresolved. Lung cancers develop uncommonly in both dogs and cats. As no predisposing cause has been established for these neoplasms, a viral cause could potentially be important.

Mammary gland neoplasia is common in humans, dogs, and cats. A role of PVs in human breast cancer was first suggested in 1992 when high-risk HPV DNA was detected in 5 of 17 neoplasms [15]. Since then, other studies have reported HPV DNA within 0%–46% of epithelial breast cancers [16]. Papillomavirus DNA has generally been more frequently detected in neoplastic than non-neoplastic samples [16], although what, if any, role PVs play in promoting cancer development is uncertain.

Papillomavirus DNA has been detected in 0%–80% of human bladder cancers. While the marked variability in the results makes interpretation difficult, HPV DNA was detected in less than 5% of the urothelial cancer samples in around half of the studies [17]. While some studies reported differences in the rate of detection between neoplastic and non-neoplastic bladder samples, the role of HPVs in human bladder cancer is currently unresolved. Within the veterinary species, PVs, along with exposure to bracken fern, have been associated with bladder cancer in cattle [18]. Bladder cancers are observed rarely in both dogs and cats, and no cause has been determined in either species.

In humans, some evidence suggests that PV infection could cause epithelial cancers of the lung, breast, and bladder. To the authors’ knowledge, whether PV DNA is detectable in lung, mammary gland, or bladder cancer in dogs and cats has never been investigated. If PV DNA is detectable in these cancers, this would suggest a potential role of PVs in cancer development, potentially allowing for the development of novel prevention strategies. Additionally, detecting PV DNA in epithelial neoplasms of the lung, mammary gland, or bladder of dogs or cats would provide comparative evidence supporting the role of these viruses in the development of these cancers in humans.

## 2. Materials and Methods 

Cases of canine and feline epithelial lung, mammary gland, and bladder cancer were identified by searching necropsy submissions between 2013 and 2018 to the School of Veterinary Science at Massey University. The original diagnosis was confirmed using hematoxylin and eosin-stained sections. These sections were also examined for the presence of histological features suggestive of PV infection. Thick sections of formalin-fixed paraffin-embedded tissue blocks were cut using a new microtome blade and mounted on slides. A new scalpel blade was then used to scrape a sample of the primary neoplasm off the slide for DNA extraction.

Total DNA was extracted as previously described [19]. The presence of amplifiable DNA within the canine and feline samples was confirmed by amplifying a portion of the *glyceraldehyde-3-phosphate dehydrogenase (GAPDH)* and the *p53* gene, respectively [20,21]. Three consensus primers were used to amplify PV DNA from the samples. The MY09/MY11 and FAP59/FAP64 primers amplify a section of the PV *L1* gene, while the CP4/5 primers amplify a section of the *E1* gene [22,23,24]. These three primers were used as they have all been reported to amplify a broad spectrum of cutaneous and mucosal PVs from humans and non-human species. The reaction conditions for these primers were the same as previously described [19,25]. Positive controls for the FAP59/64 primers included DNA extracted from a canine papilloma that contained *Canis familiaris* papillomavirus (CPV) type 2 and DNA extracted from a feline BISC that contained *Felis catus* papillomavirus (FcaPV) type 2. Positive controls for the MY09/11 primers included DNA from the same canine papilloma and a feline BISC that contained FcaPV-3. DNA extracted from a canine oral SCC that contained CPV-17 and a feline oral papilloma that contained FcaPV-1 were used as positive controls for the CP4/5 primers. The negative controls did not contain template DNA. Amplified DNA was purified and sequenced as previously described, and it was compared to known sequences in GenBank using the basic local alignment search tool (http://www.ncbi.nlm.nih.gov/GenBank).

## 3. Results

A total of 47 canine samples and 25 feline samples were included in the study. The samples of canine tumors included 14 pulmonary adenocarcinomas, 20 mammary gland adenocarcinomas, and 13 transitional cell carcinomas (TCCs), while the feline tumors comprised 10 pulmonary adenocarcinomas, 10 mammary adenocarcinomas, and 5 TCCs, as shown in Table 1. Careful evaluation of all the samples did not reveal histological features suggestive of PV infection.

By using the GAPDH or p53 primers, the presence of amplifiable DNA was confirmed in all the samples except one feline pulmonary adenocarcinoma and one feline mammary adenocarcinoma. The FAP59/64, MY09/11, and CP4/5 primers amplified DNA from the positive control tissues as expected. Small amounts of PV DNA were amplified from two canine mammary adenocarcinomas using the FAP59/64 primers. As there was insufficient amplified DNA for sequencing, 1 µL of the purified amplicon was then used as the template in an additional PCR reaction using the FAP59/64 primers. This additional step amplified sufficient DNA for sequencing. Sequencing of the two ~400bp amplicons revealed that one was 98% similar to bovine papillomavirus (BPV) type 2, while the other was 98% similar to HPV-2. The small number of mismatches were considered most likely due to errors during the sequencing process rather than the identification of different strains of PV. Total DNA was re-extracted from the formalin-fixed paraffin-embedded blocks of both samples and small quantities of the same sequences were again amplified by the FAP59/64 primers. Using the CP4/5 primers, a small quantity of PV DNA was amplified from a canine TCC. Again, amplification of the purified amplicon was required to allow sequencing, which revealed 97% similarity to HPV-2. However, the CP4/5 primers did not amplify this sequence from DNA re-extracted from the same tissue block. Examination of the three samples that contained PV DNA by a pathologist (JSM) boarded by the American College of Veterinary Pathologists did not reveal any unique histological features. No PV DNA was amplified from any sample using the MY09/11 primers. Papillomaviral DNA was not amplified from any of the lung cancers or any of the feline mammary gland or bladder cancers.

## 4. Discussion

Detecting PV DNA within a cancer may suggest PV-influenced cancer development. In the present study, no CPV or FcaPV DNA was amplified from any of the 70 cancers that were able to be evaluated. In many PV-induced neoplasms in other species, evidence that a PV may be involved in cancer development can also be derived from the observation that PV-induced lesions can progress to neoplasia. Such progression is seen in cats where pre-neoplastic BISCs, which often show histological evidence of PV infection, such as blue-grey cytoplasmic inclusions and koilocytosis [26], progress to invasive SCCs [27,28]. Likewise, in dogs, invasive carcinomas, albeit rarely, have developed as a progression from a cutaneous viral plaque or oral papilloma [29,30,31,32]. To the authors’ knowledge, no PV-induced hyperplastic or pre-neoplastic lesions have been reported in the lung, mammary gland, or bladder of dogs or cats, and no such lesions were observed in the present study. Furthermore, the presence of cells displaying histological evidence of PV infection were not visible in these tissues in dogs or cats. Therefore, considering the lack of detection of PV DNA in these cancers, and the absence of any histological evidence of PV infection within the lesions, PVs do not appear to be a significant cause of lung, mammary gland, or bladder cancers in dogs and cats. 

As PV DNA has been detected in many studies of human lung, breast, and bladder cancers, the failure to detect either CPV or FcaPV DNA in any of the 70 samples in the present study was perhaps surprising. However, large variations in the rate of detection of PV DNA have been reported in studies of human cancers, with a proportion of studies of lung, breast, and bladder cancers not detecting PV DNA within the cancers [13,14]. It is possible that the preponderance of studies that report high rates of PV DNA within these neoplasms is due to publication bias, whereby studies in which HPV DNA was detected are easier to publish than studies that confirm previous negative results. This suggests that it is possible that the ‘true’ rate of HPV infection in human lung, breast, and bladder cancer could be low. Therefore, the failure to detect CPV or FcaPV DNA within the samples in the present study would not be completely inconsistent with studies of these cancers in humans. 

For a PV to cause cancer, the virus has to be able to infect the epithelial component of a tissue. Infection by PVs is currently thought to be dependent on infectious virus particles in the environment coming into contact with, and infecting, basal cells [33]. While it is possible that PVs may be able to enter the mammary gland during lactation, it is harder to explain how PVs could come into contact with the epithelium of the bladder. Inhalation of PV particles is a plausible explanation of infection of the lungs by PVs, although loss of the mucous blanket would presumably be required to allow access to the respiratory epithelial cells. While evidence from cattle and humans suggest that PVs can circulate in blood cells [34], it is unknown whether or not an infectious PV particle can escape the blood vessel and infect the epithelial surface. An additional consideration is that PVs, presumably because their life-cycles are intimately coordinated with epithelial maturation, tend to have a very limited tissue tropism in the body. While it is feasible that PVs could infect the stratified squamous epithelium of the teat sinus, it is less certain whether or not they could infect the cuboidal epithelial lining in smaller mammary gland ducts, the pseudostratified respiratory epithelium, or the stratified transitional epithelium lining the bladder. The failure to detect any CPV or FcaPV DNA within the 70 samples in the present study suggests not only that PVs are unlikely to cause cancer in these tissues but also that tissues are not asymptomatically infected by PVs. 

While the results of the present study are most consistent with PVs not causing these cancers in dogs and cats, it is possible that the tissues could have been infected by an undetected PV type. Currently, there are 19 different CPV types and five different FcaPV types [5,8]. By using the three sets of consensus primers, all 24 known PV types are expected to be amplified [8,35]. However, there are undoubtedly additional PV types that infect dogs and cats. Furthermore, it appears likely that at least some of these additional PVs have remained undetected, because the commonly used consensus primers do not amplify their DNA. The possibility that some of these cancers are caused by currently undetected PV types cannot be excluded. 

Geographical differences in the rate of detection of HPVs within lung, breast, and bladder neoplasms have been reported. For example, while lung cancers in Canada, Singapore, and the Netherlands did not contain HPV DNA, in Brazil, Korea, Greece, and Taiwan [14], HPV DNA was detected in over 40% of lung cancers. In the present study, only samples from dogs and cats from New Zealand were evaluated. Therefore, it remains possible that geographical differences in the rate of PV detection could also exist for these cancers in dogs and cats. Alternatively, different subtypes of neoplasms may be more likely to contain HPV DNA. For example, HPV DNA was detected in around 40% of squamous cell carcinomas of the lung, but less than 10% of large cell carcinomas [36]. Within bladder cancers, the highest rates of HPV DNA detection were in highly anaplastic neoplasms and neoplasms with squamous differentiation [37]. In contrast, the histological grade or subtype of breast cancer in humans has not been associated with significantly different rates of HPV detection [38]. In the present study, the evaluated samples represented samples of multiple different grades and types. It is therefore possible that by focusing on particular subtypes of lung, mammary gland, or bladder neoplasms, a higher rate of PV infection may have been detected. 

Immunosuppression is a well-established risk factor for PV-induced neoplasms of the genitals and skin in humans [39]. In contrast, there is no evidence to suggest that immunosuppression is associated with an increased rate of detection of PV DNA in cancers of the lung, breast, and bladder. This absence of an association between immunosuppression and the development of PV-associated lung, breast, and bladder cancers suggests that HPV may not be a significant cause of these cancers. In the present study, it was unknown if any of the animals were immunosuppressed, and it is possible that by targeting neoplasms that develop in immunosuppressed animals, a higher rate of PV DNA may have been detected.

Bladder cancers have previously been associated with PV infection in cattle [18]. These cancers are restricted to cattle that ingest bracken fern [15]. Bracken fern contains carcinogens and ingestion has been reported to cause bladder cancer in rats, presumably without the involvement of PVs [40]. However, bracken fern also causes immunosuppression that could potentially allow greater infection of the bladder by PVs without the PVs causing the bladder neoplasia [3]. A significant difference between human bladder cancers that contained HPV and the bovine cancers is that the bladder cancers that develop in cattle are derived from the transitional epithelium lining of the bladder, as well as the mesenchymal tissues within the bladder wall [41]. Bladder cancers in cattle are associated with infection by BPV-2, which is classified as a deltaPV [18]. The deltaPVs are unique, because, as demonstrated by deltaPV-induced equine sarcoids, they can infect non-host species and cause mesenchymal cell proliferation [42]. The unique properties of BPV-2 may explain why this PV type has been consistently associated with bladder cancers in cattle, while bladder cancers in humans, dogs, and cats often do not contain PV DNA.

The first round of amplification revealed the presence of PV DNA within two canine mammary gland neoplasms and one canine TCC. One of the mammary gland samples contained BPV-2 DNA, while both other amplicons were from HPV-2. While cross-species infection of horses by BPV-2 is well recognized [42], this PV has never previously been reported to infect dogs. The HPVs are considered strictly host-specific. Therefore, all three PVs were considered likely to be contaminants. The possibility that these were contaminants was supported by the very small quantities of DNA amplified from the samples. All three samples were re-extracted and HPV-2 was no longer amplified from the TCC sample. This suggests that the contamination had occurred during initial DNA extraction. In humans, HPV-2 is a common cause of non-genital cutaneous papillomas (warts), and DNA most likely contaminated the sample from an infected person. The BPV-2 and HPV-2 sequences remained amplifiable when DNA was re-extracted from these tissue blocks. This suggests that the samples were contaminated when the tissues were initially taken during post-mortem examination. This contamination was probably from a cow that had been previously necropsied in the same room for BPV-2, and from a person who had performed the necropsy for HPV-2. The possibility that these PVs contributed to neoplasm development cannot be excluded, but examination of these lesions histologically did not reveal any features consistent with PV infection, which supported the PV DNA being present as contamination. These results are a reminder of the potential for the detection of contaminants when using highly sensitive PCR methods. It is interesting to speculate that if HPV had been detected from a sample of human cancer, it would have been more likely to be interpreted as significant, rather than a likely contaminant as in this case. 

## 5. Conclusions

In conclusion, the results of this study do not support a role of PVs in the development of epithelial neoplasms of the lung, mammary gland, or bladder in dogs or cats. Furthermore, the absence of any detectable DNA within the samples suggests that PVs do not commonly infect the epithelium of the lung, mammary gland, or bladder. However, the possibility that undetectable PV types are present within the samples cannot be excluded.

## Figures and Tables

**Table 1 animals-09-00668-t001:** Summary of the neoplasms included in this study. PV is papillomavirus, BPV-2 is bovine papillomavirus type 2, and HPV-2 is human papillomavirus type 2. * indicates that the DNA from these PVs was probably the result of contamination.

Neoplasm type	Total Number	Total with PV DNA (PV Type)
Canine	47	3
Pulmonary adenocarcinoma	14	0
Mammary gland	20	2
Simple carcinoma	11	1 (BPV-2) *
Intraductular papillary carcinoma	4	1 (HPV-2) *
Mixed mammary carcinoma	3	0
Complex carcinoma	2	0
Transitional cell carcinoma	13	1 (HPV-2) *
Feline	23	0
Pulmonary adenocarcinoma	9	0
Mammary gland	9	0
Simple carcinoma	9	0
Transitional cell carcinoma	5	0

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
