# Peer review of "Papillomavirus DNA is not Amplifiable from Bladder, Lung, or Mammary Gland Cancers in Dogs or Cats"

_animals, 2019, doi:10.3390/ani9090668_

Round 1
Reviewer 1 Report
This paper is very well-written and does not require any major editorial type-editing.
The manuscript is well organized and the data is presented in an understandable manner. The study design was sound and authors justified findings. Thorough explanations for results were provided and results were not over-interpreted. Discussion section is very thorough.
No major edits are required.
Author Response
Please see the revised version attached.

Reviewer 2 Report
Broad comments
The subject of the article "Papillomavirus DNA is not amplifiable from bladder, lung, or mammary gland cancers in dogs or cats" fits the aims and scope of Animals. The subject of the study is interesting and important for the development of veterinary oncology. All sections: Introduction, Material and methods, Results and Discussion are well and interestingly written. The article describes negative results, which makes it relatively short and simple. However, the subject of the study (infections in PVs in cancer) is important and the article provides valuable information useful both for scientists and clinicians. Despite the negative results, the topic is interesting and fully described. Due to the found contamination with human virus, it also indicates new possibilities of interpretation of existing data in human medicine.
Specific comments
Abstract
Page: 1 Line:23 - please change breast to mammary
Page: 1 Line 26-27 - "DNA from a bovine PV type was amplified from one sample with DNA from a human PV type amplified from two samples." - incomprehensible sentence
Author Response
Thank you for the positive feedback.
Page: 1 Line:23 - please change breast to mammary
This has been changed as suggested (line 23)
Page: 1 Line 26-27 - "DNA from a bovine PV type was amplified from one sample with DNA from a human PV type amplified from two samples." - incomprehensible sentence
This sentence has been re-written to improve clarify (line 28)
Reviewer 3 Report
see attached

Author Response
Thank you for your positive feedback.
The word 'epithelial' has been added throughout the manuscript as suggested to ensure readers are aware that mesenchymal neoplasms are not being discussed.
The last sentence of the abstract has been changed as suggested (lines 17 and 30).
The contribution of a ACVP-boarded pathologist has been added as suggested (line 131).
A table summarizing the samples included in this study has been added as suggested. The table also includes the results of the PCR analysis to make this information more simple of the reader.